# Signaling of Tumor-Derived sEV Impacts Melanoma Progression

**DOI:** 10.3390/ijms21145066

**Published:** 2020-07-17

**Authors:** Aneta Zebrowska, Piotr Widlak, Theresa Whiteside, Monika Pietrowska

**Affiliations:** 1Maria Sklodowska-Curie National Research Institute of Oncology, Gliwice Branch, 44-100 Gliwice, Poland; aneta7zebrowska@gmail.com (A.Z.); piotr.widlak@io.gliwice.pl (P.W.); 2UPMC Hillman Cancer Center, University of Pittsburgh, Pittsburgh, PA 15213, USA; whitesidetl@upmc.edu; 3Department of Pathology, University of Pittsburgh School of Medicine Pittsburgh, Pittsburgh, PA 15261, USA

**Keywords:** small extracellular vesicles (sEV), tumor-derived exosomes (TEX), melanoma cell-derived exosomes (MTEX), proteomics, tumor microenvironment, biomarkers

## Abstract

Small extracellular vesicles (sEV or exosomes) are nanovesicles (30–150 nm) released both in vivo and in vitro by most cell types. Tumor cells produce sEV called TEX and disperse them throughout all body fluids. TEX contain a cargo of proteins, lipids, and RNA that is similar but not identical to that of the “parent” producer cell (i.e., the cargo of exosomes released by melanoma cells is similar but not identical to exosomes released by melanocytes), possibly due to selective endosomal packaging. TEX and their role in cancer biology have been intensively investigated largely due to the possibility that TEX might serve as key component of a “liquid tumor biopsy.” TEX are also involved in the crosstalk between cancer and immune cells and play a key role in the suppression of anti-tumor immune responses, thus contributing to the tumor progression. Most of the available information about the TEX molecular composition and functions has been gained using sEV isolated from supernatants of cancer cell lines. However, newer data linking plasma levels of TEX with cancer progression have focused attention on TEX in the patients’ peripheral circulation as potential biomarkers of cancer diagnosis, development, activity, and response to therapy. Here, we consider the molecular cargo and functions of TEX as potential biomarkers of one of the most fatal malignancies—melanoma. Studies of TEX in plasma of patients with melanoma offer the possibility of an in-depth understanding of the melanoma biology and response to immune therapies. This review features melanoma cell-derived exosomes (MTEX) with special emphasis on exosome-mediated signaling between melanoma cells and the host immune system.

## 1. Introduction

Small extracellular vesicles (sEV), also known as exosomes (EX), are virus-size (30–150 nm) membrane-bound vesicles released by different cell types under both normal and pathological conditions. They represent a subset of the heterogeneous group of extracellular vesicles (EV) that in addition to sEV include larger (250–1000 nm) microvesicles (MV, ectosomes) and the largest (>1000 nm) apoptotic bodies (AB). EV vary in size, biogenesis, release mechanisms, and biochemical properties. sEV or exosomes are formed in the endosomal network as intraluminal vesicles (ILV) within the multivesicular bodies (MVB) and are released to the extracellular space when MVBs fuse with the cellular plasma membrane. In contrast, MV are produced by outward budding (“blebbing”) of the plasma membrane, while apoptotic bodies are released when cells undergo the programmed cell death [1,2,3,4,5,6,7]. At present, inconsistency in the EV nomenclature exists causing much confusion in the field, which extends to the methodology for sEV isolation and characterization. Currently, the most common vesicle isolation methods, including ultracentrifugation, do not adequately discriminate between various EV subsets. To ease confusion, a simplified nomenclature has been recently adopted in the literature that distinguishes small EV (i.e., <200 nm) and medium/large EV (>200 nm). The class of small EV consists mostly of exosomes, yet other types of EV, e.g., small MV, could also copurify with this fraction [8,9,10,11]. In this review, the terms “exosomes” (EX) and (small) extracellular vesicles (sEV) are used interchangeably for simplicity and to stay in line with the recent guidelines from International Society for Extracellular Vesicles (ISEV) [8].

Among various subsets of EV in body fluids of cancer patients, tumor-derived exosomes, called TEX, have attracted much attention as major mediators of intercellular communication in the tumor microenvironment (TME) and as potentially promising diagnostic, prognostic, and predictive biomarkers in cancer and other diseases. The knowledge of the molecular profiles and biology of TEX offers the possibility of a deeper understanding of pathological processes involved in cancer development and may provide important clinical information about disease activity and response to treatment. Today, TEX are considered as prime candidates for a liquid tumor biopsy, and much effort is being invested in validation of this concept. In this review, we summarize recent insights into the biology and composition of melanoma cell-derived exosomes (MTEX) and provide an up-to-date account of their pleiotropic role in melanoma progression and response to anti-melanoma therapies.

## 2. General Characteristics of sEV

sEV are produced and released by various cell types, including hematopoietic cells and a broad variety of normal or malignant tissue cells [5,7,12,13]. sEV can be isolated from supernatants of cultured cells or diverse body fluids, including blood, urine, saliva, breast milk, ascites effusions, bile, tears, nasal secretions, amniotic, synovial, cerebrospinal, lymphatic, and seminal fluids [5,7,10,14,15,16,17,18,19,20,21,22,23]. The molecular content of sEV is of special interest, as it reflects the nature of parental cells. sEV originating from different cell types share their general features, such as the structure of the bilayer lipid–protein membrane and key molecular components. Their molecular cargo consists of proteins (including cytoskeletal proteins, transmembrane proteins, tetraspanins, heat shock proteins, adhesion proteins, enzymes, immunocompetent proteins e.g., death receptor ligands: tumor necrosis factor ligand (FasL, CD95L or CD178) or TNF-related apoptosis-inducing ligand (TRAIL), check-point receptor ligands such as: programmed death-ligand 1 (PD-L1), inhibitory cytokines such as: interleukin 10 (IL-10), IL-6, TNF-α, IL-1β, and TGF-β1, prostaglandin E2, major histocompatibility molecules MHC-I and II, and tumor-associated antigens), nucleic acids (including DNA, RNA, miRNA, non-coding RNA), lipids, and low-molecular-weight metabolites (including alcohols, amides, amino acids, carboxylic acids, sugars). Proteomic analysis of exosome cargos revealed that some proteins are typical for most of these vesicles (including proteins such as Rab2, Rab7, flotillin, and annexin; cytoskeletal proteins, including actin, myosin, tubulin; or heat shock proteins, such as Hsc70 and Hsc90). Tetraspanins, such as CD9, CD63, CD81, CD83, along with housekeeping proteins, ALIX (programmed cell death 6-interacting protein) and TSG101 (tumor susceptibility gene 101 protein), are widely considered as exosome markers. However, in addition to the “common” set of components, sEV of different cellular origins may also carry proteins that are cell-type specific [8,9,10,11,12,24]. It has been shown in many studies that the molecular profile of TEX is distinct from that of sEV derived from non-malignant cells such as dendritic cells (DC), T cells, and others [6,7,11,12,13,25]. However, it should be noted that the discrimination of TEX from other types of sEV in patients’ plasma using, e.g., tetraspanins as sEV-specific markers has been limited and currently, separation of TEX from total sEV in plasma has not been readily available or reliably performed.

sEV circulating in body fluids represent a complex mixture of vesicles released by many different cell types. The majority of studies of TEX present in body fluids of cancer patients are based on analyzes performed with a mixture of sEV derived from different normal or pathological cells. Separation of TEX from this mixture remains a challenge due to the lack of universal cancer-specific antigens that could be targeted for TEX isolation. Nevertheless, a few studies that used specific membrane markers for isolation of TEX from body fluids have been reported. Chondroitin sulfate proteoglycan 4 (CSPG4) was used for separation of melanoma-derived TEX, referred to as MTEX, from vesicles released by non-malignant cells [26,27,28]. Glypican 1 (GPC1) was used to isolate TEX from plasma of patients with pancreatic cancer [29], and prostate-specific membrane antigen (PSMA) was used to isolate TEX from plasma of prostate cancer patients [30]. CD34 antigen, a unique marker of AML blasts, was used to isolate TEX from the plasma of patients with acute myeloid leukemia [31]. Moreover, MAGE3/6 antigen was used to identify TEX present in sera of patients with melanoma or head and neck squamous cell carcinoma (HNSCC) [25]. In contrast to plasma-derived sEV, TEX isolated from supernatants of cancer cell lines are putatively homogenous. These TEX derived from tumor cell lines are an excellent in vitro model for investigations of interactions of TEX with other cells. In fact, much of what is known about TEX signaling, uptake by responder cells, and reprogramming in TME is derived from in vitro and in vivo studies of TEX isolated from supernatants of tumor cell lines.

sEV have gained interest due to their essential role in “normal” intercellular signaling and communication, which impact the physiological balance and homeostasis as well as disease progression. [1,6,32,33,34,35,36]. Importantly, sEV can modulate the phenotype/functions of recipient cells, even those located in distant organs [2,3,35,36,37,38]. Moreover, the role of TEX in cancer progression has been reported for many cancer types, including ovarian, prostate, breast, lung, colorectal and gastric cancers, melanoma, and acute myeloid leukemia [26,37,38,39,40,41,42,43,44]. TEX are being intensively investigated because they play a key role in the reorganization of the TME, remodeling functions of the cells residing in the TME, and enhancing their contribution to tumorigenesis, metastasis, cancer immune escape, as well as resistance to cancer treatment [34,44,45,46,47,48,49,50,51,52,53,54]. The potential role of TEX in cancer biology is schematically illustrated in Figure 1. A better understanding of the mechanisms underlying TEX-mediated reprogramming of normal cells in TME is expected to be clinically significant, leading to improved cancer diagnosis/prognosis and treatment. In addition, TEX are considered to be an attractive source of cancer biomarkers.

## 3. General Features of Melanoma

Melanoma is the most aggressive skin cancer whose incidence has been increasing worldwide. Melanoma prognosis is generally poor, as it has a high potential for vascular invasion, metastasis, and recurrence [55,56,57]. Primary melanomas detected in an early stage and completely removed surgically show favorable outcome, with 5-year disease-specific survival rates of 99% [58]. However, melanoma cells tend to metastasize to distant sites, most often to lungs and brain, while evading the host immune system. Hence, the survival rate dramatically decreases when the cancer metastasizes, and 1-year survival rate drops to 35%–62% [57,59], while 5-year survival rate drops to 25% [58]. Currently, surgery is the treatment of choice for patients with cutaneous melanoma, and the adjuvant treatment scheme is usually tailored individually. Melanoma is sensitive to immune checkpoint inhibitors, such as anti-CTLA4 (anti-cytotoxic T-lymphocyte-associated protein 4 or anti-CD 152) and anti-PD1 (anti-programmed cell death protein 1 or anti-CD279) monoclonal antibodies (mAb), and to small-molecule targeted drugs, such as serine/threonine-protein kinase B-Raf (BRAF) or mitogen-activated protein kinase (MEK) inhibitors. Hence, different treatment schemes, including radiotherapy and/or adjuvant treatments with anti-BRAF/MEK inhibitors and anti-PD-1 mAb are currently used, depending on the patient’s clinical situation [60]. Despite these novel combinatorial therapies, tumor escape from the immune control and development of primary or acquired therapy resistance that occurs in about half of melanoma patients remain the major therapeutic barrier [61,62,63]. Melanoma cells communicate with other cells present in the TME, including components of the immune system, via melanoma cell-derived exosomes (MTEX). Hence, in-depth knowledge of MTEX composition and function is expected to bring better understanding of the mechanisms determining the response of melanoma to treatments.

## 4. The Molecular Cargo of MTEX

### 4.1. The Proteome of MTEX

Recently, studies of proteomic profiles of cancer-derived sEV have been much intensified [64,65,66,67]. However, only limited data are available about the proteome of melanoma-derived TEX. A great part of the available literature focuses on in vitro studies with TEX isolated from supernatants of various melanoma cell lines [68,69,70,71,72,73]. Ex vivo studies performed with TEX isolated from the blood of melanoma patients are rare [74]. It is important to emphasize that only sEV derived from melanoma cell lines are “pure” MTEX, as those present in the plasma will be mixtures of sEV derived from many different cells. The available studies of MTEX have utilized different proteomic approaches. Most of them are based on shotgun LC-MS/MS strategies (i.e., tryptic digestion of proteins, followed by nano HPLC-MS/MS analysis of the resulting peptides), while others are based on LC-MS/MS analysis of proteins separated by 1D or 2D SDS-PAGE [75]. Currently available proteomic studies demonstrate differences in protein profiles of MTEX in comparison to melanocyte-derived sEV [68,69]. In addition, MTEX derived from melanoma cell lines with a different tumorigenic potential appeared to have distinct proteomic profiles [70]. Moreover, proteomic analysis of sEV present in the plasma of melanoma patients and healthy donors showed clear differences and revealed increased levels of TYRP2, VLA-4, and HSP70 in patients’ samples [74]. However, the knowledge of the proteome in TEX produced by melanoma cells remains rather limited, and the available data are difficult to compare because they represent distinct experimental models. A summary of data on proteomics profiling of MTEX released by melanoma cell lines is presented in Table 1.

### 4.2. Micro RNA Component of MTEX

Micro RNAs are small (19–25 nucleotides) non-coding RNAs that play an important role as regulators of cell differentiation, metabolism, proliferation, or innate and adaptive immunity [76,77,78]. Micro RNA profiles of TEX differ from miRNA profiles of their donor cancer cells as well as from profiles of sEV released by normal cells [78,79,80,81]. The majority of available works report miRNA signatures of pure MTEX released in vitro by melanoma cell lines [69,82,83,84,85,86,87,88,89,90]. Moreover, a few studies addressed miRNA composition of sEV derived from serum/plasma of melanoma patients [82,91,92,93,94]. Similar to proteomics data, few consistencies were observed among these studies due to different models applied. Nevertheless, there were 6 MTEX-upregulated miRNA species reported in more than one study: miR-494 [82,83], let-7c [69,84], miR-690 [84,85], miR-17 [84,91], and miR-494 [82,83], while miR-125b was reported to be downregulated in MTEX or sEV from plasma of melanoma patients [83,92]. Noteworthy, all the above mentioned miRs are known to be involved in cancer cell invasion, migration, and proliferation as well as in inflammatory processes linked to tumorigenesis and cancer progression [82,83,84,85,86,91,92].

Xiao et al. showed significant differences in miRNA content of exosomes isolated from normal melanocytes and malignant cell lines (HEMa-LP and A375), respectively [69]. In this study, 130 miRNAs were upregulated and 98 miRNAs downregulated in MTEX versus melanocyte-derived EX. The majority of differently expressed miRNAs were associated with tumor aggressiveness, including fifteen miRNAs known to be associated with melanoma metastasis: miR-138, miR-125b, miR-130a, miR-34a, miR-196a, miR-199a-3p, miR-25, miR-27a, miR-200b, miR-23b, miR-146a, miR-613, miR-205, miR-149, let-7c [69]. Another study reported enrichment of miRNA-494, which is known for its high metastatic potential, in MTEX released by A375 cells. A series of functional experiments performed by Li et al. demonstrated that intercellular transport of miR-494 in MTEX was responsible for melanoma metastasis [82]. Blocking of exosomal transfer of miR-494 by a knockdown (KO) of Rab27a induced cellular apoptosis and inhibited tumor growth and metastasis in vitro and in human xenografts [82].

Another analysis of the miRs upregulated in sEV of patients with metastatic melanoma (miR-17-5p, miR-19a-3p, miR-149-5p, miR-21, and miR-126-3p) focused on discovery of putative targets of these miRNAs [91]. Among their targets were genes associated with skin response to UV irradiation, genes coding the tumor protein p53 (TP53)/retinoblastoma protein (RB1) and genes related to the TGF-β/SMAD pathway. Upregulation of miRNAs controlling TP53/RB1 activation and the TGF-β/SMAD signaling pathway might play an important role in melanoma progression, as the TGFβ/SMAD pathway regulates the G1/S checkpoint in normal melanocytes [91]. Moreover, miR-17 was identified as a potential oncomiR not only in melanoma but also in other malignancies [93,94]. Association of miR-19a upregulation with increased melanoma invasiveness was confirmed by Levy et al. [95]. Upregulation of miR-21 and miR-19a is associated with increased proliferation, low apoptosis, invasiveness, and high metastatic potential, as reported for various human tumor cells [96,97], while KO of miR-21 in B16 melanoma cells reduced their metastatic potential [98,99]. The oncogenic properties of miR-21 may be a result of down-regulation of the tumor suppressors: PTEN, PDCD4 and the antiproliferative protein BTG2. In addition, miR-21 induced the IFN pathway with protumorigenic effects [98,99]. High abundance of another oncomiR—miR-1246 was detected in MTEX isolated from patient-derived melanoma cell lines, namely, DMBC9, -10, -11, and -12 [83,100]. Many other studies have confirmed its high concentration in sEV from the plasma of patients with various cancers, including melanoma [101,102]. The overexpression of miR-222 in MTEX and cells is also associated with tumor initiation, differentiation, increased cell motility, and invasion, as well as cancer progression [87]. MiR-222 inhibits anti-neoplastic functions of p27, CDKN1B, and c-Fos by down-modulation of their gene expression, reduces apoptosis, and allows proliferation by induction of the PI3K/AKT pathway [89]. Müller et al. showed the importance of let-7a in melanoma development [103]. Let-7a regulates the expression of integrin β-3, the promotor of melanoma progression. The loss of let-7a expression in MTEX derived from 8 different melanoma cell lines resulted in higher integrin β-3 levels in melanoma cells, enhancing their migratory and invasive potential [103]. Finally, let-7a was detected in serum EX as a factor differentiating stage I melanoma patients from non-melanoma subjects [104]. Altogether, the literature supports the key role played by miRs transferred by melanoma-associated TEX in oncogenesis and melanoma metastasis.

In addition to microRNA, MTEX contain mRNA transcripts of genes expressed in melanoma. Sets of mRNAs with different abundance in MTEX and in melanosome-derived exosomes were identified, including 945 transcripts associated with cancer and 364 associated with dermatological diseases [69]. Among upregulated transcripts there was DNA topoisomerase I (TOP1), which is known to be associated with aggressive, advanced tumors and poor prognosis in melanoma [69,105]. Among downregulated transcripts there were ATP-binding cassette, sub-family B, member 5 (ABCB5), which activates the NF-κB pathway enhancing p65 protein stability [106] and is also known to be closely co-regulated with melanoma tumor antigen p97 (tumor growth regulator—melanotransferrin, MTf) [107], and TYRP1 encoding tyrosinase-related protein 1, which is considered as an inhibitor of TYRP1-dependent miR-16 mediating tumor suppression [108,109].

## 5. Biological Activity of MTEX

The multi-level contribution of MTEX to tumorigenesis accounts for activation of biological processes enabling cancer immune evasion, as well as molecular and metabolic remodeling of tumor micro- and macro-environment, favoring cancer growth and metastasis. The in-depth knowledge of the pleiotropic role of MTEX in the natural history of melanoma has a great potential clinical application in the disease diagnosis, treatment design, and prognosis of patient’s outcomes. MTEX are involved in a plethora of functions involved in initiation, progression, and metastasis of tumors, which is schematically depicted in Figure 2 (according to [26,37,73,74,83,84,85,86,87,88,89,90,91,92,93,94,95,96,97,98,99,100,101,102,103,104,105,106,107,108,109,110,111,112,113,114,115,116,117,118,119,120,121,122,123,124,125,126,127,128,129,130,131,132,133,134,135,136,137,138,139,140,141,142,143,144,145,146,147,148,149,150]). The most essential functions of MTEX are addressed more specifically in the following sub-chapters.

### 5.1. MTEX Participate in the Reprogramming of Immune Cells

Growth and progression of cancer involve the escape from the immune surveillance as the sine qua non condition. Emerging evidence supports the idea that MTEX are involved in facilitating tumor escape from the host immune system [33,44,47,49,110,111,112,113,114,115,116,117]. However, in most of the studies reported in this context, melanoma cell lines were used as a source of MTEX. Düchler et al. showed that cancer-induced immunosuppression was mediated by MTEX, and involved an antigen-specific mechanism [118]. The authors provided evidence that MTEX transferred MHC class I receptor proteins from cancer cells to the surface of antigen-presenting cells (APC). At the same time, CD86 and CD40 (co-stimulatory molecules required for differentiation and proliferation of T cells) were down-regulated, while the production of immunosuppressive cytokine IL-6 was induced. Collaboration of TGF-β transported by MTEX was also demonstrated in this process. The authors hypothesized that MTEX-mediated transfer of the combination of TAA-derived peptide-MHC complexes with immunosuppressive cytokines was a part of antigen-specific tolerance induction enabling melanoma immune escape [118]. The mechanism of melanoma immune escape is also related to the suppression of T cell functions. This can be attributed to an increased level of PD-L1 in MTEX. This immunosuppression is driven by the interaction between PD-L1 carried by MTEX and PD-1 receptors on CD8^+^ T cells, leading to inhibition of T-cell functions [111,119]. MTEX are also enriched in Fas ligand (FasL) and APO2 ligand (APO2L)/TRAIL, both known as inducing factors of T cell-apoptosis [120]. Another possible mechanism for the suppression of T cell function by MTEX is through the upregulation of PTPN11 protein, which was found to negatively regulate interferon, IL-2, and T cell receptor signaling pathways [121]. Wu et al. confirmed that B16F0-derived MTEX are enriched with Ptpn11 mRNA and can increase PTPN11 dose-dependently in recipient cells. In addition to upregulating PTPN11 in lymphocytes, MTEX derived from B16F0 locally suppressed responses of cells to IL-12 (anti-tumor immunity enhancer) via inhibition of IL12RB2 expression in primary CD8^+^ T cells. These inhibitory mechanisms of the immune cell response to IL-12 are complemented by B16F0 release of the Wnt-inducible signaling protein 1 (WISP1) that blocks T cell response to IL-12 [121,122]. Furthermore, the cargo of MTEX might alter mitochondrial respiration of cytotoxic T cells and up-regulate genes associated with the Notch signaling pathway [84]. Immunosuppressive activity of MTEX depends on their ligands that engage the T cell receptor (TCR) and IL-2 receptor (IL-2R). Recent studies showed that MTEX inhibited signaling and proliferation of activated primary CD8^+^ T cells, inducing their apoptosis [25,32,90]. Furthermore, MTEX significantly promoted conversion of CD4(+) T cells to CD4(+)CD25(+)FOXP3(+) T regulatory cells (Treg) enhancing their suppressor functions [25]. Vignard et al. additionally confirmed decreased TCR signaling in T cells as a result of the enrichment in miRNAs regulating TCR signaling and TNF-α secretion (miR-3187-3p, miR-498, miR-122, miR149, miR-181a/b) in MTEX [90].

Accumulating evidence reveals that TNF is negatively regulated by miR-21. This may explain the effects of miR-21 on cell proliferation, migration, invasion, and transformation associated with excessive miR-21 levels in MTEX. Moreover, some results suggest that TNF can promote miR-21 biogenesis [123] as well as the turnover of PDCD4 in macrophages [124]. Yang et al. also showed that increased levels of miR-21 associated with a decreased level of TNF were consistent with elevated IL-10 protein expression and increased Arg1 macrophage expression, which could explain poor immune responses against cancer cells [98]. On the other hand, Fabri et al. reported that miR-21 which was found to be enriched in MTEX might also act as a ligand by binding to receptors of the Toll-like receptor (TLR) family members, murine TLR7, and human TLR8, in immune cells. Triggering the TLR-mediated prometastatic inflammatory response in responder cells might promote tumor growth and metastasis [125].

Stimulation of TLR2^+^ DC by tumor-derived TLR2 ligands was reported to drive inhibitory signals leading to dysfunctional activity of DC in murine melanoma [126]. Modulation of immune response by MTEX was confirmed by Zhou et al. [85]. They observed that B16-derived MTEX induced apoptosis of CD4^+^ T cells in vitro and promoted the growth of tumor cells implanted in mice. The opposite results were reported by blocking MTEX release (disrupting the expression of Rab27a), thus confirming the proposed mechanism. Further, they showed that B16-derived MTEX induced activation of caspase-3, caspase-7, and caspase-9, reducing the level of anti-apoptotic proteins, such as BCL-2, BCL-xL, and MCL-1 in CD4^+^ T cells [85]. MTEX can also alter the functions of natural killer (NK) cells. They were found to modulate the tumor immune responses by inhibiting the cytotoxic activity of NKs and downregulating the expression of NKG2D, NKp30, NKP46, and NKG2C proteins on the surface of NK cells [26,42,127].

### 5.2. MTEX Participate in the Reprogramming of TME

TME plays a major role in cancer growth and evolution. Diverse cells such as fibroblasts, endothelial, epithelial, and mesenchymal cells or immune cells present in the TME might be reprogrammed by MTEX to favor tumor growth [45,56]. Accumulating data provide evidence that MTEX promote epithelial-to-mesenchymal transition (EMT), which promotes metastasis [46,48,55,57,128]. The mitogen-activated protein kinase (MAPK) signaling pathway is activated during the MTEX-mediated EMT, with the involvement of Let-7i, a miRNA modulator of EMT [104]. Furthermore, acquisition of the EMT-like phenotype is enforced by expression of other key regulators of EMT induction, including ZEB2 and Snail 2 [119,129]. Upregulation of ZEB2 and Snail 2 in primary melanocytes after co-culture with MTEX was confirmed by Xiao et al. This process was accompanied by decreased expression of E-cadherin and increased expression of vimentin [104]. The interplay between MTEX and myeloid stem cells (MSCs) induce the emergence of a tumor-like phenotype with PD-1 and mTOR overexpression in naïve MSCs in vitro and fast tumor progression in vivo [119]. Interaction networks build basing on genes overexpressed in recipient cells upon co-incubation with MTEX identified a variety of other exosomal molecules, apart from PD-1 and mTOR, which might affect tumor progressions, such as MET, Ras, RAF1, Mek, ERK1/2, MITF, BCL2, PI3K, Akt, KIT, JAK STAT3, or ETS1 [119].

MTEX transform fibroblasts into proangiogenic cancer-associated fibroblasts (CAF) in vitro and in vivo. CAF are known to support development of pre-cancerous micro- and macro-environments [86,130,131]. Zhao et al. discovered that incubation of MTEX with fibroblasts resulted in a significant increase of VCAM-1 expression, and this enhancement was even stronger when EX were derived from highly metastatic melanoma cells [131]. Overexpression of miR-155 in MTEX was found to be the trigger factor for the proangiogenic switch of fibroblasts into CAF [86]. MTEX-mediated delivery of miR-155 to fibroblasts suppressed expression of cytokine signaling 1 (SOCS1), that activates the JAK2/STAT3 signaling pathway which, in turn, regulates the expression of proangiogenic factors. Elevated expression of vascular endothelial growth factor A (VEGFa), fibroblast growth factor 2 (FGF2), and matrix metalloproteinase 9 (MMP9) in fibroblasts after incubation with MTEX was confirmed in this study [86]. Shu et al. also reported the presence of exosomal miR-155 and miR-210 across six melanoma cell lines [89] and showed that miRNA cargo of MTEX was capable of reprogramming the metabolism of human adult dermal fibroblasts (HADF). In this study, miR-155 upregulated glucose metabolism (i.e., increased glycolysis), while miR-210 decreased oxidative phosphorylation under non-hypoxic conditions. Exposure of HADF to MTEX resulted in upregulated aerobic glycolysis and downregulated oxidative phosphorylation in stromal fibroblasts, with consequently increasing extracellular acidification [89]. Furthermore, the acidic environment led to upregulation of over 50% of exosomal proteins involved in cancer progression in MTEX derived from the primary non-tumorigenic MEL501 cell line [73]. The upregulated proteins were associated with focal adhesion, actin cytoskeleton regulation, leukocyte trans-endothelial migration, regulation and modification of cell morphology, HSP family proteins, proteoglycans related to cancer, small GTPase mediated signal transduction, and epidermal growth factor receptor (EGFR) signaling pathways [73]. This shows that MTEX are important contributors to changes in the TME that are responsible for creating favorable conditions for the pre-metastatic niche. On the one hand accelerated aerobic glycolysis ensures more effective energy production, but on the other hand, the acidic microenvironment drives immune suppression and creates a pro-metastatic environment [73,89,132].

The pro-angiogenic effects of MTEX are well-documented. MTEX cargos are enriched in pro-angiogenic cytokines, including IL-1α, FGF, GCS-F, TNFα, leptin, TGFα, and VEGF [107]. MTEX also mediate the transfer of miR-9 from melanoma to endothelial cells (EC), which triggers the JAK-STAT pathway and enhances the migratory propensity of vascular cells as well as the formation of a tumor-supporting vascular net [133]. Additionally, it was reported that increased WNT5A signaling, which is known to promote melanoma metastasis, induced a Ca^2+^-dependent release of exosomes containing the pro-angiogenic VEGF and MMP2 factors in melanoma cells [134].

### 5.3. MTEX Can Modulate Tumor Progression and Invasiveness

In general, TEX may induce tumor growth in vitro and in vivo [135,136]. It was reported that B16BL6-derived MTEX induced proliferation and inhibited apoptosis of murine melanoma B16BL6 cells, while inhibition of MTEX release by the N-Smase inhibitor suppressed melanoma growth. Noteworthy, the uptake of MTEX resulted in increased levels of cyclin D1, p-Akt (cell proliferation-related proteins), Bcl-2 (survival-related protein), and decreased level of Bax (apoptosis-related proteins) [137]. Peinado et al. reported that the oncoprotein MET selectively enriched in MTEX released by metastatic melanoma cells promoted the tumorigenic potential of melanoma [74]. Pre-conditioning of bone marrow (BM) with MTEX obtained from a highly metastatic melanoma B16-F10 cell line promoted mobilization of bone marrow-derived cells (BMDC), increasing tumor vasculogenesis, invasion, and metastasis. Comparative analysis of the protein content in MTEX from highly metastatic and poorly metastatic melanoma cells confirmed MET signaling as the principal mediator of BM progenitor cell “education”. Pre-treatment of BM cells with B16-F10 MTEX resulted in HGF-induced S6 and ERK phosphorylation compared to non-treated controls. Effectors of MET-mediated BM progenitor cell mobilization, i.e., S6-kinase (mTOR pathway) and ERK (MAPK pathway), are known mediators of HGF/MET signaling. Further, the metastatic spread and organotropism of highly metastatic B16-F10 primary tumors were reduced by the BM of mice “educated” with the low-metastatic B16-F1 MTEX that lacked the MET receptor. These data suggested that non-metastatic MTEX might educate the BM and prevent metastatic disease, a finding that is worth further exploration. Finally, it was confirmed that MET expression was elevated in sEV circulating in the plasma of patients with metastatic melanoma [74]. Additionally, influence of metabotropic glutamate receptor 1 (GRM1) expressed on melanoma cells was tested for cell migration and invasiveness [138]. This neuronal receptor induces in vitro melanocytic transformation and spontaneous malignant melanoma development in vivo. Moreover, modulation (decrease) of GRM1 expression results in a decrease in both cell proliferation in vitro and tumor progression in vivo. Isola et al. verified a hypothesis that exosomes released by a GRM1-positive (metastatic) cell line made GRM1-negative (non-metastatic) cells acquire features characteristic for GRM1-positive cells, i.e., to migrate, invade, form colonies, and exhibit anchorage-independent cell growth. They found that acquiring these features was not connected with expression of this receptor on GRM1-negative cells. Another aspect of the potential role of MTEX in tumorigenesis is analysis of specific RAB genes involved in sEV secretion (RAB1A, RAB5B, RAB7, RAB27A) [74]. Rab27a is a regulator of protein trafficking and melanoma proliferation [139]. Reduced expression of Rab27a resulted in decreased sEV production, and in decreased release of pro-angiogenic factors (PlGF-2, osteopontin, and PDGF-AA) from tumor cells, interfering with BMDC mobilization and tumor invasiveness [74]. These results are in line with the latest findings of Guo and colleagues [140], who reported that the GTPase RAB27A was overexpressed in melanoma patients and correlated with poor patient survival. A loss of RAB27A expression in melanoma cell lines blocked invasion and cell motility in vitro, and spontaneous metastasis in vivo. Furthermore, RAB27A-expressing MTEX promoted the invasion phenotype of melanoma cells in contrast to MTEX without RAB27A [140]. All in all, these results suggest that RAB27A promotes the biogenesis of a distinct pro-invasive MTEX subpopulation [74,140].

MTEX are also involved in preparation of metastatic niche for melanoma in lymph nodes and lungs and in reprogramming of innate osteotropism of melanoma cells [74,141,142]. MTEX from a highly-metastatic B16-F10 cell line promoted lymph nodes (LN) metastasis in mice [142] and were detected after 24h in the interstitium of the lung, BM, liver, and spleen (organotropic sites for B16-F10 metastasis), but not in the circulatory system [74]. Several genes responsible for the recruitment of melanoma cells (stabilin 1, ephrin receptor β4, and αv integrin), extracellular matrix remodeling (Mapk14, uPA, laminin 5, Col 18α1, G-α13, p38), vascular growth (TNF-α, TNF-αip2, VEGF-B, HIF-1α, Thbs1) [142], and effectors of pre-metastatic niche formation such as S100A8, S100A9 [74] were upregulated by B16-F10 MTEX. The osteotropism of melanoma cells is related to the activation of the SDF-1/CXCR4/CXCR7 axis. MTEX were found to promote osteotropism of not-osteotropic melanoma cells (SK-Mel28, WM266) in vitro through membrane CXCR7 up-regulation. Thus, MTEX were found to contribute to bone metastasis in melanoma [141].

## 6. MTEX as Potential Clinical Biomarkers

MTEX present in body fluids of melanoma patients are a promising source of prognostic biomarkers as a new type of so-called liquid biopsy. Alegre et al. performed an analysis of the established melanoma biomarkers such as: MIA, S100B, and tyrosinase-related protein 2 (TYRP2) in sEV isolated from sera of stage IV melanoma patients, patients with no evidence of disease (NED), and healthy donors (HD) [37]. The levels of MIA and S100B were significantly higher in melanoma patients in comparison to HD and NED patients. Furthermore, patients with high EV concentration of MIA had shorter median survival compared to those with lower MIA levels (4 versus 11 months; *p* < 0.05). Hence, the data suggest the potential diagnostic and prognostic utility of MIA in plasma sEV [37]. Levels of MIA, along with growth/differentiation factor 15 precursor protein (GDF15) showed a significant increase in the whole secretome of uveal melanoma versus non-malignant cells [143], which was in line with the results of Alegre et al. [37]. Tenga et al. showed that miR-532-5p and miR-106b present in serum sEV could be used for classification of melanoma patients, including differentiation of patients with metastatic and non-metastatic disease and stage I-II patients from stage III-IV patients [144]. In addition, miR-17, miR-19a, miR-21, miR-126, and miR-149 were found to be expressed at significantly higher levels in patients with metastatic sporadic melanoma compared to familial melanoma patients or healthy controls [91]. On the other hand, levels of miR-125b in sEV were significantly lower in patients with advanced melanoma compared with disease-free patients with melanoma and healthy controls, while there was no statistical difference in the miR-125b levels between patients and controls when analyzing serum samples [92].

Melanoma is sensitive to immune checkpoint inhibitors (such as anti-CTLA4 and anti-PD1 monoclonal antibodies) and small-molecule targeted drugs (such as BRAF inhibitors and MEK inhibitors). However, many patients with melanoma fail to respond to these therapies, and the mechanisms of resistance to a therapy are not understood [61,62,63,145,146]. The accumulating data suggest the importance of MTEX in understanding these mechanisms and the role of MTEX as predictive biomarkers of response to immune therapies and outcome [55,56,57,147]. Higher levels of miR-497-5p in circulating sEV during MAPKi-based therapy of cutaneous metastatic melanoma patients (with BRAFV600 mutations) were significantly correlated with progression-free survival (hazard ratio of 0.27) [147]. Increased level of miR-497-5p was also associated with prolonged post-recurrence survival in resected metastases from patients with metastatic III (lymph nodes) and metastatic IV cutaneous malignant melanoma (CMM) [148]. Treatment with vemurafenib and dabrafenib induced miR-211-5p up-regulation in melanoma-derived EV, both in vitro and in vivo, thus promoting survival in parent melanoma cells despite a down-regulation of pERK1/2 by BRAF inhibitors [146]. What is more, transfection of miR-211 in low-expressing miR-211–5p melanoma cells resulted in enhanced proliferation of melanoma cells. What is more, 100-fold increase in miR-211–5p expression in vemurafenib-treated miR-211-5p-transfected cells was found with no reduction of cells proliferation upon BRAF inhibitor treatment. These findings suggest that miR-211-5p up-regulation upon vemurafenib treatment allows these cells to survive and grow into a population of cells that have reduced sensitivity to vemurafenib. Going further, inhibition of miR-211-5p in a vemurafenib resistant cell line decreased cell proliferation. The outcome of the study of Lunavat et al. leads to better understanding of possible mechanisms of acquiring by patients’ resistance to the BRAF inhibitors treatment by showing that miR-211-5p can reduce the sensitivity to vemurafenib treatment in melanoma cells by regulating cellular proliferation. [146]. Another group of “new drugs” used in the treatment of melanoma are immune checkpoint inhibitors. Anti-PD-1 antibodies are frequently used in melanoma treatment to rejuvenate anti-tumor immunity, and in the majority of patients the response is durable, yet not all melanoma patients respond to this therapy [60,149]. Chen et al. reported positive correlation between exosomal-PD-L1 (Exo-PD-L1) level and IFN-γ, both in vitro using melanoma cell lines and in vivo in patients with metastatic melanoma [111]. Upregulation of PD-L1 by IFN-γ in metastatic melanoma leads to functional suppression of CD8+ T effector cells enabling melanoma growth and metastasis. In part, this explains low response rate to anti-PD-1 therapy (pembrolizumab). The level of circulating Exo-PD-L1 distinguished clinical responders from non-responders to pembrolizumab treatment. Since the level of exosomal PD-L1 was altered early during the anti-PD-1 therapy, the authors suggest that it might be an indicator of response to treatment [111]. A recent paper by Cordonnier et al. describes monitoring of circulating Exo-PD-L1 in melanoma patients treated with immune checkpoint inhibitors and BRAF/MEK inhibitors. This prospective clinical study confirmed a significantly higher level of Exo-PD-L1 in plasma of melanoma patients compared to soluble PD-L1 and demonstrated that the level of Exo-PD-L1 inversely correlated with patients’ response to therapy [150]. The results of this clinical study provide a rationale for monitoring Exo-PD-L1 level as a potential predictor of the melanoma patients’ response to treatment and outcome [150].

Clinical relevance of MTEX-based biomarkers is currently limited by the necessity of separation of MTEX from other fractions of sEV circulating in body fluids. Recently, however, the anti-CSPG4 mAb was used for the separation of MTEX and sEV produced by normal tissue from the plasma of melanoma patients [26,27,28]. CSPG4^+^ MTEX captured from the plasma of melanoma patients are highly enriched in melanoma-associated antigens (MAA) in comparison to CSPG4(-) non-MTEX, including CSPG4, TYRP2, MelanA, Gp100, VLA4. Moreover, several immunostimulatory (CD40, CD40L, CD80, OX40, OX40L) and immunosuppressive (PDL-1, CD39, CD73, FasL, LAP-TGFβ, TRAIL, CTLA-4) proteins were enriched in MTEX compared to sEV purified from plasma of healthy donors [26,28]. Noteworthy, looking at individual differences among proteins in the cargo of MTEX and non-MTEX, significant correlations with disease activity were observed for both fractions of vesicles. For example, non-MTEX ability to induce apoptosis of T cells positively correlated with the disease stage [28]. The obtained data suggest that features of both MTEX and non-MTEX, as well as individual MTEX/total sEV ratios, might be useful for monitoring melanoma progression [26,28]. In addition to CSPG4, other melanoma-specific or enhanced proteins might also be considered as potential markers of MTEX. This includes several melanoma-associated antigens (MAA-4, MAA-B2, and melanoma antigen recognized by T-cells) found in MTEX released by 7 different melanoma cell lines with various phenotypic features (non-tumorigenic, tumorigenic, metastatic) [70]. Moreover, several other cancer-related proteins (NRAS, Src, c-Met, c-Kit, EGFR, MCAM, annexin A1, HAPLN1, LGALS1, GALS3, NT5E, and PMEL) were detected in MTEX originating from various melanoma cell lines [69,70]. Therefore, several candidates for MTEX-markers are known that could be used for the immune capture of MTEX circulating in the body fluids of melanoma patients. Hence, the emerging concept of MTEX-based biomarkers of melanoma will meet the necessary methodological support in the nearest future.

## 7. Future Directions

Although the number of publications reporting on sEV in melanoma is growing exponentially, the resulting knowledge remains limited. Most likely, this is due to several factors that impede research of sEV. First, no uniformly accepted nomenclature for EV has been established, creating havoc in the definition of investigated EV. Further, no standardized procedures for the isolation of different EV types exist, leading to differences in contamination levels and co-isolation of various vesicles. The criteria and methods of EV characterization are also not clear and seem to change as we progress in the understanding of the EV heterogeneity. Despite the recommendations updated yearly by the International Society of Extracellular Vesicles (ISEV), published papers often provide incomplete data creating further confusion. The emerging view of the complex biology of EV requires strict criteria for the definition of phenotypes, genotypes, and functions of participating EV. Specifically, in a large body of available data on melanoma-associated sEV in body fluids, their origin is often unclear. Until recently, melanoma cell lines had been the only reliable source of MTEX. However, research performed with vesicles produced by cell lines does not adequately reflect interactions taking place in body fluids or tissues. Separation of MTEX from plasma by immune capture allowed for a more relevant evaluation of their characteristics and functions in disease and comparisons of data between individual patients. While this represents considerable progress, ex vivo analysis of MTEX also provides only a limited view of their biological agenda in the TME and the periphery. In vivo studies of MTEX in murine models of melanoma are critical for the translation of signaling mediated by MTEX in vitro to cells, tissues, and organs in animals. Correlative studies of MTEX and clinical endpoints in melanoma progression, resistance, or response to therapies are growing in numbers and the concept of MTEX as a liquid tumor biopsy is slowly crystallizing. Understanding of multicellular MTEX-mediated signaling and their reprogramming activities in the TME opens a way for the use of MTEX-induced changes as yet another biomarker of disease activity. The next step is to develop and implement reliable means for the isolation and molecular characterization of MTEX from body fluids and tissues of patients with melanoma. At present, these methods are in the discovery stage, and the emerging results are promising not only due to successful subsetting of sEV into MTEX and non-MTEX, but also because of evidence that mechanistic and functional studies of MTEX can yield new and previously unsuspected information. For example, the ability of MTEX to simultaneously deliver to recipient cells multiple and often contradictory, i.e., suppressive vs. stimulatory signals have alerted us to the possibility of regulatory functions MTEX might exercise in vivo. Similarly, the realization that MTEX utilize surface proteins as well as miRs to transmit signals to recipient cells alerts us to ask why these two signaling pathways co-exist and how they impact the biology. As melanoma biomarkers, MTEX might provide a more reliable diagnostic, prognostic, or outcome data than total sEV isolated from body fluids. Future validation studies encompassing all aspects of MTEX isolation, characterization, and signaling will be necessary to move the field forward and translate current knowledge to clinically applicable strategies and methods. In this respect, antibody-based microarrays, multiparameter quantitative flow cytometry, and targeted proteomics are emerging as the tools applicable to serial monitoring of MTEX in body fluids of patients with melanoma. The future will likely see numerous such studies performed as part of clinical trials designed to validate the roles of MTEX in the biology of melanoma.

## Figures and Tables

**Figure 1 ijms-21-05066-f001:**
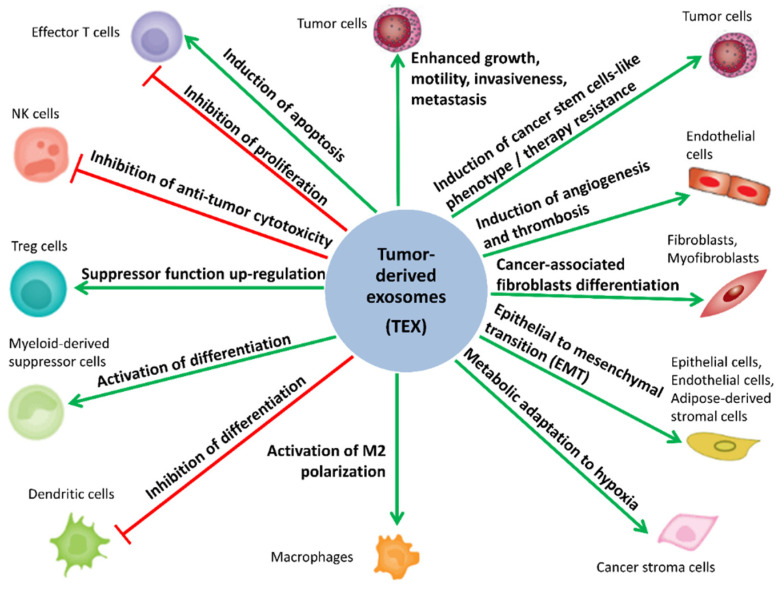
Role of tumor-derived small extracellular vesicles (sEV) (TEX) in cancer biology. Tumor-derived exosomes (TEX) are involved in intercellular signaling and communication between tumor cells and non-malignant cells residing in the tumor microenvironment. TEX reprogram these cells to acquire functions favoring tumor growth and metastasis. TEX-induced changes include enhancing cancer immune escape, remodeling of the tumor stroma, molecular and metabolic reprogramming, and promotion of angiogenesis. Green arrows indicate the processes stimulated by TEX. Red lines with blunt ends indicate processes inhibited by TEX.

**Figure 2 ijms-21-05066-f002:**
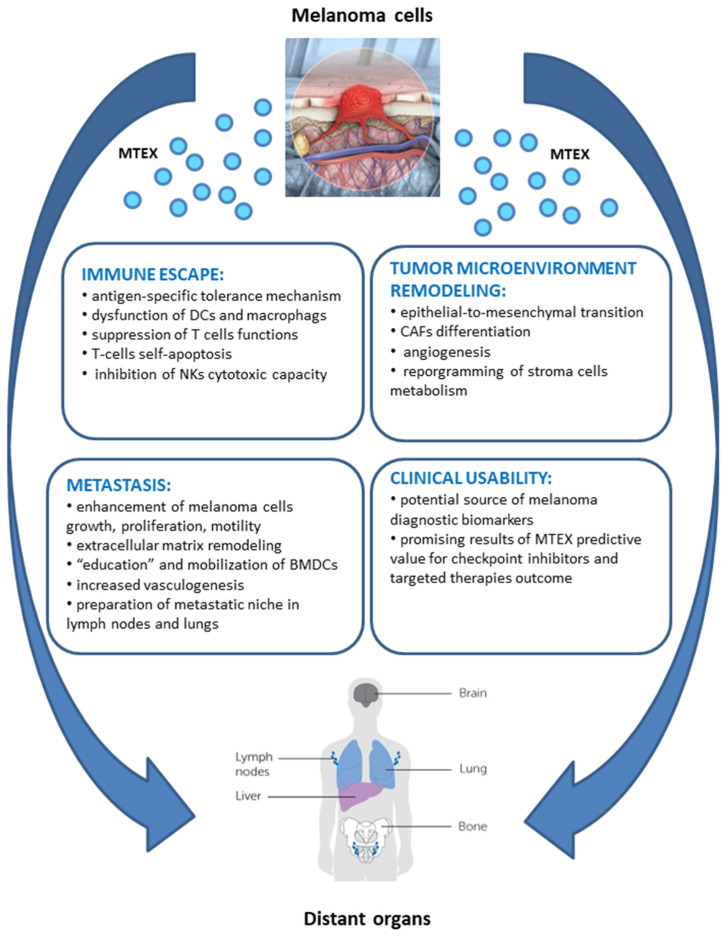
Pleiotropic effect of MTEX on melanoma biology.

**Table 1 ijms-21-05066-t001:** Proteomics Profiling of Melanoma Cell-Derived Exosomes (MTEX) Released in Vitro by Melanoma Cell Lines.

Cell Line	Method of MTEX Purification and Characterization	MS Approach	Major Findings	Ref.
MeWo, SK-MEL-28 (human)	UC/TEM, WB, 1D/2D SDS-PAGE	MALDI-TOF MS/MS	A few proteins identified in MTEX for the first time: prostaglandin regulatory-like protein (PGRL), p120 catenin, syntaxin-binding proteins 1 and 2, septin 2 (Nedd5), ezrin, radixin, tryptophan/aspartic acid (WD) repeat-containing protein 1	[68]
A375 (human)	UC/TEM, NTA, WB	LC-MS/MS	Different sets of proteins present in MTEX and melanocyte-derived EV, including annexin A1, HAPLN1, GRP78, endoplasmin precursor (gp 96), TUBA1B, PYGB), ferritin, heavy polypeptide 1 (MTEX-upregulated), annexin A2, syntenin-1, MFGE8, OXCT (MTEX-downregulated)	[69]
MNT-1, G1, 501 mel, SKMEL28, Daju, A375M, 1205Lu (human)	UC+SEC/WB, TEM, NTA	nanoLC-MS/MS	Different sets of proteins present in MTEX from nontumorigenic, tumorigenic, and metastatic cell lines, including EGFR, PTK2/FAK1, EPHB2, SRC, LGALS1/LEG1, LGALS3/LEG3, NT5E/5NTD-CD73, NRAS, KIT, MCAM/MUC18, MET specific for metastatic cell lines	[70]
B16-F1 (murine)	UC+SEC/CEM, DLS, IA-FCM	uHPLC-MS	10 most abundant proteins: CD81, CD9, histones (H2A, H2B, H3.1, H4), heat shock proteins (HSPA5/GRP78, HSC71), syntetin-1	[71]
B16-F10 (murine)	UC+SEC, UF+SEC/TEM, NTA, WB	nanoLC-MS/MS	Different sets of proteins identified in low- and high-density MTEX, including ACTN4 and CCNY enriched in LD-MTEX and EPHA2 enriched in HD-MTEX	[72]
Mel501 (human)	UC+SEC/WB, CM	RPLC-MS/MS	Different sets of proteins identified in MTEX released in neutral and acidic environment (pH 6.7 and 6.0, respectively), including HRAS, NRAS, TIMP3, HSP90AB1, HSP90B1, HSPAIL, HSPA5, GANAB, gelsolin, and cofilin upregulated in acidic conditions.	[73]

Abbreviations: methods of EX purification: UC—ultracentrifugation, UF—ultrafiltration, SEC—size-exclusion chromatography; methods of EX characterization: NTA—nanoparticle tracking analysis, DLS—dynamic light scattering, TEM—transmission electron microscopy, CEM—cryo-electron microscopy, CM—confocal microscopy, WB—Western blotting.

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
