# Peer review of "Signaling of Tumor-Derived sEV Impacts Melanoma Progression"

_ijms, 2020, doi:10.3390/ijms21145066_

Round 1
Reviewer 1 Report
This a well-written comprehensive review by Zebrowska et al. on the current state of knowledge on tumor-extracellular vesicles in general and a more focused discussion on their role and implications for melanoma.
A few minor comments:
r435-437: "These finding ... treatment [146]". This sentence does not seem correct and may need some restructuring.
The authors refer to low response rate to anti-PD-1 treatment in melanoma, while in fact, melanoma is actually one of the better responding cancer types to this type of treatment. I do not really understand why the authors are so negative on this.
r450: "soluble PD-L" should probably read "soluble PD-L1".
Author Response
Dear Reviewer,
thank you for your evaluation of our manuscript ijms-868267. Enclosed is the revised manuscript, corrected according to your comments. All introduced changes are highlighted in yellow. A point-by-point response to specific questions is given below.
Q1. r435-437: "These finding ... treatment [146]". This sentence does not seem correct and may need some restructuring.
A1. The sentence has been restructured and some additional information has been included in the revised manuscript. The amended text is given below:
What is more, transfection of miR-211 in low-expressing miR-211–5p melanoma cells resulted in enhanced proliferation of melanoma cells. What is more, 100-fold increase in miR-211–5p expression in vemurafenib-treated miR-211–5p–transfected cells was found with no reduction of cells proliferation upon BRAF inhibitor treatment. These findings suggest that miR-211–5p up-regulation upon vemurafenib treatment allows these cells to survive and grow into a population of cells that have reduced sensitivity to vemurafenib. Going further, inhibition of miR-211-5p in a vemurafenib resistant cell line decreased cell proliferation. The outcome of the study of Lunavat et al. leads to better understanding of possible mechanisms of acquiring by patients' resistance to the BRAF inhibitors treatment by showing that miR-211–5p can reduce the sensitivity to vemurafenib treatment in melanoma cells by regulating cellular proliferation [147].
Q2. The authors refer to low response rate to anti-PD-1 treatment in melanoma, while in fact, melanoma is actually one of the better responding cancer types to this type of treatment. I do not really understand why the authors are so negative on this.
A2. Thank you very much for this comment, as it was not our intention to sound negative on anti-PD-1 treatment in melanoma. Yes, the fact is that in the majority of patients the response is durable, and we added this fact to the text in the revised manuscript. Our intention was to show that Exo-PD-L1 may have a potential to be a helpful biomarker for even better, personalized treatment design. The amended sentence is given below:
Anti-PD-1 antibodies are frequently used in melanoma treatment to rejuvenate anti-tumor immunity, and in the majority of patients the response is durable, yet not all melanoma patients respond to this therapy [60, 150].
Q3. r450: "soluble PD-L" should probably read "soluble PD-L1".
A3. Yes, we corrected it in text for soluble PD-L1.
We would like to emphasize that we are very grateful for all your efforts to improve our manuscript. We hope that the current version of the text meets your expectations.
Faithfully yours,
Corresponding Author – Monika Pietrowska
Reviewer 2 Report
Signaling of Tumor Derived sEV impacts Melanoma Progression
In this well written manuscript by Zebrowska and colleagues reviewed the role of small extracellular vesicles/exosomes in melanoma progression and its utilization in the clinic. They explore the complexities of defining exosomes and inconsistencies in defining what exosomes are. They provided a systematic overview of exosome literatures and emphases on the importance in understanding the cargo of melanoma exosomes in order to not only understand the mechanisms by which melanoma exosomes’ involvement in melanoma progression but also the identification of biomarkers for treatment response and the uses for diagnosis and prognosis. The authors may consider the following comments:
Comments:
- In line 18 when the authors stated that TEX contains cargo similar but not identical to that of parent producer cells, do you mean that the tumor derived exosomes have similar but not identical cargo (protein, lipids and RNA) than the normal counterpart exosomes released by untransformed cells? (i.e. the cargo of melanoma exosomes is similar but not identical to melanocyte exosomes?) If so, please make this clearer.
- Line 52 you wrote ISEV, please define this acronym.
- Line 73 what do you mean by immunocompetent proteins? Please define in the text.
- In figure 1 please define acronyms: CSC and CAFs.
- According to the American Cancer Society, the 2020 melanoma statistics for the 5-year survival rate for melanoma is 99% and after metastasis its 25%. (Siegel, R.L., Miller, K.D. and Jemal, A. (2020), Cancer statistics, 2020. CA A Cancer J Clin, 70: 7-30. doi:3322/caac.21590)
- In line 142 you wrote MTEX please define this acronym.
- In figure 2 please define MCs.
- Line 431, please define CMM.
- It has been shown in vitro that human GRM1 positive melanoma metastatic cells released exosomes enhanced the migratory, invasiveness and colony forming abilities of recipient cells when compared to GRM1 negative cells. The authors may consider including this publication in section 5.3. of the review.
Isola, A.L., et al., Exosomes released by metabotropic glutamate receptor 1 (GRM1) expressing melanoma cells increase cell migration and invasiveness. 2017. 9(1).
Author Response
Dear Reviewer,
thank you for your evaluation of our manuscript ijms-868267. Enclosed is the revised manuscript, corrected according to your comments. All introduced changes are highlighted in yellow. A point-by-point response to specific questions is given below.
Q1. In line 18 when the authors stated that TEX contains cargo similar but not identical to that of parent producer cells, do you mean that the tumor derived exosomes have similar but not identical cargo (protein, lipids and RNA) than the normal counterpart exosomes released by untransformed cells? (i.e. the cargo of melanoma exosomes is similar but not identical to melanocyte exosomes?) If so, please make this clearer.
A1. Yes, that’s what we meant. We corrected it accordingly in the revised manuscript:
TEX contain a cargo of proteins, lipids, and RNA that is similar but not identical to that of the “parent” producer cell (i.e. the cargo of exosomes released by melanoma cells is similar but not identical to exosomes released by melanocytes), possibly due to selective endosomal packaging.
Q2. Line 52 you wrote ISEV, please define this acronym.
A2. According to the reviewer’s suggestion we defined the ISEV acronym as International Society for Extracellular Vesicles.
Q3. Line 73 what do you mean by immunocompetent proteins? Please define in the text.
A3. According to the reviewer suggestion we added examples of immunocompetent proteins in text:
(…) immunocompetent proteins e.g. death receptor ligands: FasL or TRAIL, check-point receptor ligands such as: PD-L1, inhibitory cytokines such as: IL-10, IL-6, TNF-α, IL-1ß and TGF-ß1, prostaglandin E2, major histocompatibility molecules MHC-I and II, and tumor-associated antigens) (…)
Q4. In figure 1 please define acronyms: CSC and CAFs.
A4. According to the reviewer’s suggestion we defined acronyms for CSC (Cancer-Stem Cells) and CAFs (Cancer-Associated Fibroblasts) in Figure 1.
Q5. According to the American Cancer Society, the 2020 melanoma statistics for the 5-year survival rate for melanoma is 99% and after metastasis its 25%. (Siegel, R.L., Miller, K.D. and Jemal, A. (2020), Cancer statistics, 2020. CA A Cancer J Clin, 70: 7-30. doi:3322/caac.21590).
A5. According to the reviewer’s suggestion we corrected melanoma statistics with the publication of Siegel, R.L., Miller, K.D. and Jemal, A. (2020), Cancer statistics, 2020. CA A Cancer J Clin, 70: 7-30. doi:3322/caac.21590. Therefore, the reference [58] was changed form the paper of Aubuchon et al. (in the first version of the manuscript) to the paper of Siegel et al. (in the revised text).
Q6. In line 142 you wrote MTEX please define this acronym.
A6. According to the reviewer’s suggestion we defined the acronym MTEX as Melanoma Cell-Derived Exosomes.
Q7. In figure 2 please define MCs.
A7. For better clarity we removed the acronym MCs (meaning: melanoma cells) from Figure 2 and used the term: melanoma cells, instead.
Q8. Line 431, please define CMM.
A8. According to the reviewer’s suggestion we defined the acronym CMM as Cutaneous Malignant Melanoma.
Q9. It has been shown in vitro that human GRM1 positive melanoma metastatic cells released exosomes enhanced the migratory, invasiveness and colony forming abilities of recipient cells when compared to GRM1 negative cells. The authors may consider including this publication in section 5.3. of the review. Isola, A.L., et al., Exosomes released by metabotropic glutamate receptor 1 (GRM1) expressing melanoma cells increase cell migration and invasiveness. 2017. 9(1).
A9. Thank you very much for this suggestion. We included the publication of Isola et al. in section 5.3 of the revised manuscript with the reference of [139]. The following description has also been added:
Additionally, influence of metabotropic glutamate receptor 1 (GRM1) expressed on melanoma cells was tested for cell migration and invasiveness [139]. This neuronal receptor induces in vitro melanocytic transformation and spontaneous malignant melanoma development in vivo. Moreover, modulation (decrease) of GRM1 expression results in a decrease in both cell proliferation in vitro and tumor progression in vivo. Isola et al. verified an hypothesis that exosomes released by a GRM1-positive (metastatic) cell line made GRM1-negative (non-metastatic) cells acquire features characteristic for GRM1-positive cells, i.e. to migrate, invade, form colonies and exhibit anchorage-independent cell growth. They found that acquiring these features was not connected with expression of this receptor on GRM1-negative cells.
We would like to emphasize that we are very grateful for all your efforts to improve our manuscript. We hope that the current version of the text meets your expectations.
Faithfully yours,
Corresponding Author - Monika Pietrowska